# Impact of the COVID-19 Pandemic on Patients with Gastrointestinal Cancer Undergoing Active Cancer Treatment in an Ambulatory Therapy Center: The Patients’ Perspective

**DOI:** 10.3390/healthcare9121688

**Published:** 2021-12-06

**Authors:** Koichi Taira, Hisashi Nagahara, Hiroaki Tanaka, Akie Kimura, Akinobu Nakata, Yasuhito Iseki, Tatsunari Fukuoka, Masatsune Shibutani, Takahiro Toyokawa, Shigeru Lee, Kazuya Muguruma, Masaichi Ohira, Tomoya Kawaguchi, Yasuhiro Fujiwara

**Affiliations:** 1Department of Gastroenterology, Graduate School of Medicine, Osaka City University, Osaka 545-8585, Japan; kimura.akie@med.osaka-cu.ac.jp (A.K.); m2072594@med.osaka-cu.ac.jp (A.N.); yasu@med.osaka-cu.ac.jp (Y.F.); 2Department of Gastroenterological Surgery, Graduate School of Medicine, Osaka City University, Osaka 545-8585, Japan; hisashi@med.osaka-cu.ac.jp (H.N.); hiroakitan@med.osaka-cu.ac.jp (H.T.); m2074067@med.osaka-cu.ac.jp (Y.I.); m1150909@med.osaka-cu.ac.jp (T.F.); fbxbj429@ybb.ne.jp (M.S.); t-toyokawa@med.osaka-cu.ac.jp (T.T.); m3010375@msic.med.osaka-cu.ac.jp (S.L.); m1294441@msic.med.osaka-cu.ac.jp (K.M.); masaichi@med.osaka-cu.ac.jp (M.O.); 3Department of Respiratory Medicine, Graduate School of Medicine, Osaka City University, Osaka 545-8585, Japan; kawaguchi.tomoya@med.osaka-cu.ac.jp

**Keywords:** COVID-19, gastrointestinal cancer, patient perspective

## Abstract

Background: The mortality risk increases greatly in patients with cancer if they are infected with severe acute respiratory syndrome coronavirus 2. The new American Society of Clinical Oncology (ASCO) and the European Society of Medical Oncology (ESMO) guidelines for the COVID-19 pandemic suggested modifications to the standards of care to reduce harm from treatment. However, it is unclear whether these changes suit the wishes of patients. Methods: We conducted a survey of patients with gastrointestinal cancer who were undergoing active chemotherapy in our ambulatory therapy center between 17 August and 11 September 2020. The survey comprised 18 questions on five topics: patient characteristics, lifestyle changes, disturbance in their psychological health, thoughts on the cancer treatment, and infection control in the hospital. Results: Among the 63 patients who received the questionnaire, 61 participated in the study. The COVID-19 pandemic has led to changes in their lifestyles and substantially impacted their psychological wellbeing. The incidence of anxiety and insomnia has considerably increased during the pandemic. However, female patients and patients aged 70 years or older reported no notable differences. There was no significant difference in the responses to the questions regarding thoughts on the cancer treatment. Conclusion: Our study revealed that the COVID-19 pandemic has substantially impacted patients’ lifestyles and psychological wellbeing. However, most patients preferred to continue their usual treatment without any change to their treatment plan. It is important to involve the patient in the decision-making process when formulating treatment goals.

## 1. Introduction

Coronavirus disease-19 (COVID-19) is caused by the severe acute respiratory syndrome coronavirus 2 (SARS-CoV-2) [1]. The World Health Organization (WHO) labeled COVID-19 a global pandemic on 11 March 2020 [2]. Japan reported its first case of laboratory-confirmed COVID-19 on 15 January 2020. The number of patients suffering from COVID-19 has been increasing daily since then. Therefore, to stem the spread of SARS-CoV-2, the Japanese government decided to place all of Japan under a state of emergency from 16 April to 14 May 2020 [3]. The incidence of COVID-19 decreased somewhat after that, but it began to increase again at the beginning of July 2020; the so-called second wave. By 17 August 2020, nearly 57,000 confirmed cases had been reported, with about 1000 newly diagnosed patients. Our hospital in Osaka had the second-highest number of cases. People had to take precautions to reduce their risk of contracting the infection by changing their lifestyles. WHO and the Ministry of Health, Labor and Welfare of Japan recommend the following precautions to avoid transmitting COVID-19: avoiding large events and mass gatherings, keeping a distance of at least 6 ft. from others, washing hands often, and covering the mouth and nose with a face mask [2,4]. COVID-19 is often more severe in people who are older adults or who have health conditions such as lung or heart diseases, diabetes, or conditions that affect their immune system. Cancer and cancer-related treatments frequently cause immunosuppression and the mortality risk for patients with cancer increases greatly if they are infected with SARS-CoV-2 [2,5]. The relatively higher frequency clinical visits for patients undergoing active treatment may put patients with cancer at a higher risk of being infected with SARS-CoV-2. Therefore, the new American Society of Clinical Oncology (ASCO) and the European Society of Medical Oncology (ESMO) guidelines for the COVID-19 pandemic suggested modifications to the standards of care to reduce harm from treatment [6,7,8]. However, it is unclear whether these changes would suit patients’ wishes. Although there are some reports about the mental health and preventive behavior toward COVID-19 of patients undergoing active cancer treatment, there are no reports on how the patients’ perceptions regarding their cancer treatment have changed in the wake of the COVID-19 pandemic [9,10]. Therefore, this survey aimed to clarify the impact of the COVID-19 pandemic on patients with gastrointestinal cancer undergoing active chemotherapy to compare the differences in their perspectives before and during the COVID-19 pandemic and to understand the dynamics of shared decision-making when reformulating treatment goals depending on the situation.

## 2. Materials and Methods

This was a single-center, cross-sectional study conducted via a survey at the Osaka City University Hospital in Japan. The study protocol was approved by the ethics committee of the Osaka City University Graduate School of Medicine (No. 2020-112) on 5 August 2020.

We conducted a survey on all patients with gastrointestinal cancer who were undergoing active chemotherapy in the ambulatory therapy center between 17 August and 11 September 2020. To be eligible, patients had to be 18 years of age or older. Patients who received adjuvant chemotherapy were excluded from the analysis. The survey was conducted anonymously and self-administered after distributing questionnaires that contained 18 questions on five topics: patient characteristics, lifestyle changes during the pandemic, disturbance in their psychological health, thoughts on the cancer treatment, and infection control in the hospital before and during the COVID-19 pandemic. The pre-pandemic scores were based on retrospective reporting. The survey questionnaire was developed based on a thorough review of past literature on similar topics [9,10,11,12]. Psychological characteristics were assessed based on single items rather than validated scales for research benefits, such as reduced respondent burden and ease of interpretation. The questionnaires regarding the opinions on cancer treatment were developed based on the ASCO and ESMO guidelines.

This was an exploratory study, and thus the sample size was not calculated in advance. Baseline characteristics and survey results were analyzed using descriptive statistics. The differences in the responses to these questions before and during the pandemic were analyzed for each subgroup separately. Subgroups divided by age (<70 years or ≥70 years) and by gender (male or female) were used. Regarding the 18 questions, the responses were recorded on a five-point Likert scale as: strongly inapplicable (1), slightly inapplicable (2), neither applicable nor inapplicable (3), slightly applicable (4), or extremely applicable (5). The significance of differences between groups was determined using the Wilcoxon signed-rank test. All statistical tests were two-sided, and statistical significance was set at a *p*-value of <0.05. All statistical analyses were performed using EZR (version 1.34; Saitama Medical Center, Jichi Medical University, Saitama, Japan) and the graphical user interface for R (version 3.3.2; The R Foundation for Statistical Computing, Vienna, Austria).

## 3. Results

Of the 63 patients who received the questionnaire, 61 (97%) participated in the study. A total of 50 patients, for whom a comparison between their situations before and during the pandemic could be studied, were included in this analysis. The remaining 11 patients were excluded because they did not have an answer on at least one question either before or during the COVID-19 pandemic. In total, 27 (64%) of the participants were men, and most were aged between 50 and 79 years (77%). Approximately half of the patients (42%) were 70 years or older. Colorectal cancer (48%) and gastric cancer (34%) were the most frequently diagnosed malignancies. No patient was reported to have tested positive for COVID-19. There were no patients who changed their treatment plan due to the pandemic. Patients usually talked about cancer with their attending physicians (73%) and their families (44%) (Table 1).

Among the 18 questions, a median value of 5 (extremely applicable) was recorded in response to the questions “Have you reduced the number of times you go out?”, “Do you pay attention to the control of infection?”, and “Do you often watch the news?” during the COVID-19 pandemic. On the other hand, when asking about the situation before the COVID-19 pandemic, the mean value of 5 was recorded only in response to this question, “Do you often watch the news?” (Figure 1, Table 2).

The COVID-19 pandemic caused the patients to change their lifestyles. There was a significant difference between the number of patients who reduced their number of outings and were being more careful about contracting infections before and during the COVID-19 pandemic (*p* < 0.001) (Figure 1). A significant change in lifestyle before and during the pandemic was observed in each subgroup (Table 3 and Table 4). Since the patients often watched the news both before and during the COVID-19 pandemic, there was no significant difference in this characteristic.

Regarding the toll on the patients’ psychological well-being, anxiety and insomnia were reported to have considerably increased during the pandemic (*p* < 0.01) (Table 2). However, female patients and patients aged 70 years or older did not report a significant difference in their psychological wellbeing as a result of the pandemic (Table 3 and Table 4).

On the other hand, although the patients tended to visit the hospital less frequently during the pandemic (*p* = 0.057), there was no significant difference between before and during the pandemic in the responses to the questions regarding perceptions on cancer treatment (Table 2). Further, no significant differences were found in any subgroup in this regard (Table 3 and Table 4).

Regarding questions about the infection control in hospitals, a significant difference in responses was observed in terms of whether the hospital was taking sufficient measures against infection during the pandemic when compared to the measures taken before, and the patients often looked at our hospital’s website during the pandemic (Table 2). However, there was no significant difference in the responses to these questions given by female patients or those aged 70 years or older (Table 3 and Table 4).

## 4. Discussion

According to the new ASCO and ESMO guidelines for the COVID-19 pandemic, the time cancer patients spend at the hospital should be reduced, for example, by changing intravenous regimens to oral regimens and increasing the interval between treatments. Nevertheless, a study on the patients’ perspectives regarding these recommendations is lacking. To the best of our knowledge, this is the first study to investigate patients’ perspectives on cancer treatment, specifically the views of the patients with gastrointestinal cancer undergoing active chemotherapy before and during the COVID-19 pandemic.

Our study shows the significant impact of the COVID-19 pandemic on the patients’ lifestyle and their psychological wellbeing. The patients reported a notable increase in anxiety and insomnia, a significant reduction in the number of outings, and were more careful about controlling the spread of infections during the pandemic than they were previously. However, most patients wanted to continue their treatment as usual and did not wish to change their treatment according to the recommendations of the ASCO and ESMO guidelines. The incidence of COVID-19 is lower in Japan than in other countries [13]. Therefore, few patients have been forced to change their treatment after consulting with their attending physicians. It was difficult to mitigate the risk of infection while continuing with the standard cancer treatment. There are differences in the incidence of COVID-19 between countries. Due to the cultural differences and the different national recommendations, cancer treatment protocols are different for each country during the pandemic. Therefore, it is important to involve the patient in the decision-making when formulating treatment goals depending on the country’s situation during the COVID-19 pandemic.

According to the Centers for Disease Control and Prevention (CDC) guidelines, the risk of developing severe illness after COVID-19 infection increases with age [5]. Older adults (≥70 years) had the highest risk of developing COVID-related complications. Our results did not show a significant difference in the psychological wellbeing of patients aged ≥70 years when compared to patients in younger age groups. These findings were similar to those of previous reports [14,15]. A CDC survey reported that a significantly lower percentage of the participants aged 65 years or older suffered from anxiety and depressive disorders than the participants in the younger age groups. The reasons for better mental health outcomes were considered to be sufficient precautionary measures, experience with isolation, and higher levels of wisdom [16,17,18]. However, the long-term effects of the COVID-19 pandemic on the psychological health of these participants are unclear. Older adults needed psycho-oncological support as often as younger adults. In addition, there was no significant difference in the frequency with which older participants visited the hospital website (*p* = 0.346). This finding allowed us to determine that improving our hospital website would facilitate conveying information to the patients.

Differences in reporting the degree of insomnia and in responses about sufficient measures taken by the hospital for infection control during the pandemic were observed between the sexes. Men were more likely to report insomnia and were more satisfied with the measures. Several studies have reported that the male sex is a risk factor for worse outcomes and increased mortality [19,20,21]. They may be more concerned about their health; therefore, significant differences may exist between the sexes with regard to these factors.

Our study has some limitations. First, this study analyzed a small sample of patients and was conducted at a single center. This was an exploratory study; however, retrospective calculations based on a power of 80% indicated the requirement for 45 to 72 subjects. The sample size was not too small for this study. Our study included patients with various gastrointestinal cancers from different age groups; thus, this study’s findings are able to reflect the perspectives of different patients regarding clinical practice. The regions of Japan that our center caters to are high-incidence areas of COVID-19; therefore, these findings may also be useful for other countries. Second, in our questionnaire, we included questions to compare the situation both before and during the pandemic. There was some misunderstanding when reminding the patients of this important feature of the questionnaire. However, the strength of this study was that we compared various situations before and during the COVID-19 pandemic using 18 questions. Third, the study did not include questions regarding the type of chemotherapy the patient was undergoing. All patients in this study had unresectable advanced cancers and they were receiving palliative management. We excluded those receiving adjuvant chemotherapy; hence, the patients in the study had similar thoughts to each other. Therefore, the inclusion of questions regarding chemotherapy was not considered necessary in this study. Lastly, the questionnaires were not validated. Further large studies are needed to confirm the impact of the COVID-19 pandemic.

## 5. Conclusions

The COVID-19 pandemic has affected the healthcare system in various ways. However, most patients prefer to continue their treatment as usual and do not wish to change their treatment plan according to the recommendations of guidelines. Even if the present pandemic is successfully suppressed, other infectious diseases could cause pandemics again, and other events affecting the patients’ lifestyles could unexpectedly occur. Therefore, optimal healthcare should always be delivered by formulating a management plan that accounts for the patient’s perspective.

## Figures and Tables

**Figure 1 healthcare-09-01688-f001:**
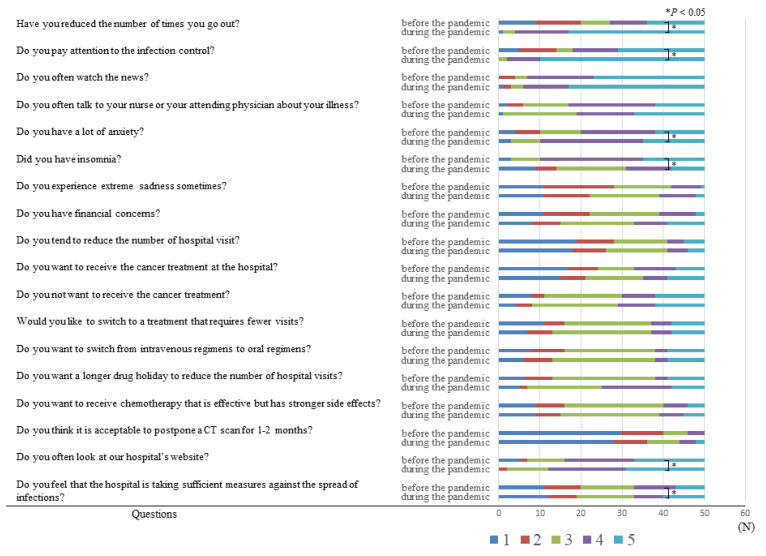
Answers to the 18 questions in the study.

**Table 1 healthcare-09-01688-t001:** Demographic characteristics.

Demographics	Options	*n*	(%)
Sex	Male	27	64
	Female	15	36
Age (years)	<40	0	0
	40–50	5	10
	50–60	10	21
	60–70	13	27
	70–80	14	29
	≧80	6	13
Type of cancer	Colorectal cancer	21	48
	Gastric cancer	15	34
	Esophageal cancer	6	14
	Small intestinal cancer	2	5
With whom they speak about their condition	Attending physician	35	73
(with overlap)	Family	21	44
	Friends	4	8
	Nurse	1	2

**Table 2 healthcare-09-01688-t002:** The distribution of the overall responses to the questions.

Questions	Before the Pandemic	During the Pandemic	*p*-Value
	Median (IQR)	Median (IQR)	
Have you reduced the number of times you go out?	3 (2–5)	5 (4–5)	<0.001
Do you pay attention to the infection control?	4 (3–5)	5 (5–5)	<0.001
Do you often watch the news?	5 (4–5)	5 (4–5)	0.444
Do you often talk to your nurse or your attending physician about your illness?	4 (3–4)	4 (3–5)	0.307
Do you have a lot of anxiety?	4 (3–4)	4 (4–5)	0.008
Did you have insomnia?	4 (3–4)	4 (4–5)	0.008
Do you experience extreme sadness sometimes?	2 (2–3)	3 (2–3)	0.08
Do you have financial concerns?	3 (2–4)	3 (2–4)	0.322
Do you tend to reduce the number of hospital visit?	3 (2–3.75)	3 (2–4)	0.057
Do you want to receive the cancer treatment at the hospital?	2 (1–3)	2 (1–3)	0.735
Do you not want to receive the cancer treatment?	3 (1–4)	3 (1–4)	0.44
Would you like to switch to a treatment that requires fewer visits?	3 (3–4)	3 (3–4)	0.371
Do you want to switch from intravenous regimens to oral regimens?	3 (2–3.75)	3 (2.25–3.75)	0.227
Do you want a longer drug holiday to reduce the number of hospital visits?	2 (2–3)	2 (2–3)	0.652
Do you want to receive chemotherapy that is effective but has stronger side effects?	4 (3–5)	4 (4–5)	0.214
Do you think it is acceptable to postpone a CT scan for 1–2 months?	3 (2–3)	3 (2–3)	0.444
Do you often look at our hospital’s website?	1 (1–2)	1 (1–3)	0.018
Do you feel that the hospital is taking sufficient measures against the spread of infections?	4 (3–5)	4 (4–5)	0.005

IQR: Interquartile Range.

**Table 3 healthcare-09-01688-t003:** The distribution of the responses by age (<70 years or >70 years) to the questions.

	<70 Years			≥70 Years		
	Before the Pandemic Median (IQR)	During the Pandemic Median (IQR)	*p*-Value	Before the Pandemic Median (IQR)	During the Pandemic Median (IQR)	*p*-Value
Have you reduced the number of times you go out?	3 (2–4)	5 (4–5)	<0.001	4.5 (2–5)	5 (4–5)	0.013
Do you pay attention to the infection control?	4 (2.75–5)	5 (5–5)	0.001	5 (3.75 –5)	5 (4.75 –5)	0.034
Do you often watch the news?	4.5 (4–5)	5 (4–5)	0.573	5 (4–5)	5 (4.75–5)	1
Do you often talk to your nurse or your attending physician about your illness?	4 (3–4)	4 (3–5)	0.246	4 (4–5)	4 (3–5)	0.681
Do you have a lot of anxiety?	4 (2.75–4)	4 (4–4.25)	0.042	4 (3–5)	4 (3.5–5)	0.174
Did you have insomnia?	3 (2–4)	3 (3–4)	0.037	2 (1–3)	3 (1–4)	0.181
Do you experience extreme sadness sometimes?	3 (2–3.5)	3 (3–4)	0.073	2 (1–3)	2 (1–3)	1
Do you have financial concerns?	3 (2.5–4)	3 (3–4)	0.265	2 (1–4)	2 (1–4)	1
Do you tend to reduce the number of hospital visit?	3 (2–3.25)	3 (2.75–4)	0.187	3 (1–3.25)	3 (1.75–4.25)	0.281
Do you want to receive the cancer treatment at the hospital?	2 (1–3)	2.5 (1–3)	1	1.5 (1–3.25)	2 (1–3.25)	0.71
Do you not want to receive the cancer treatment?	2.5 (1.75–4)	3 (2–3.25)	0.691	3 (1–4)	3 (1–4)	0.481
Would you like to switch to a treatment that requires fewer visits?	3 (2.75–4)	3 (3–4)	0.573	3 (3–5)	3.5(3–5)	0.481
Do you want to switch from intravenous regimens to oral regimens?	3 (2–4)	3 (2.75–3)	0.523	3 (2.75–3)	3 (2.00–3.25)	0.577
Do you want a longer drug holiday to reduce the number of hospital visits?	3 (2–3.25)	3 (2.75–3)	0.702	3 (1.5–4)	3 (2–5)	0.235
Do you want to receive chemotherapy that is effective but has stronger side effects?	3 (3–4)	3 (3–4)	0.425	3 (2–4)	4 (3–4.5)	0.261
Do you think it is acceptable to postpone a CT scan for 1–2 months?	3 (2–3)	3 (2–3)	0.255	3 (2–4)	3 (2–4)	1
Do you often look at our hospital’s website?	1 (1–2)	1.5 (1–3)	0.054	1 (1–2)	1 (1–2)	0.346
Do you feel that the hospital is taking sufficient measures against the spread of infections?	4 (3–5)	4 (3–5)	0.018	4 (4–5)	4 (3.75–5)	0.11

IQR: Interquartile Range.

**Table 4 healthcare-09-01688-t004:** The distribution of the responses by sex (male or female) to the questions.

	Male			Female		
	Before the Pandemic Median (IQR)	During the Pandemic Median (IQR)	*p*-Value	Before the Pandemic Median (IQR)	During the Pandemic Median (IQR)	*p*-Value
Have you reduced the number of times you go out?	3 (2–4.5)	5 (4–5)	<0.001	4 (3.5–5)	5(4.5–5)	0.034
Do you pay attention to the infection control?	4 (2–5)	5 (4.5 –5)	0.002	5(4–5)	5(5–5)	0.041
Do you often watch the news?	4 (4–5)	5 (4–5)	0.49	5(4–5)	5(4–5)	0.67
Do you often talk to your nurse or your attending physician about your illness?	4 (3.5–4)	4 (3.5–5)	0.118	4 (3–4)	3 (3–4)	0.189
Do you have a lot of anxiety?	4 (3–4)	4 (3.25–4.75)	0.086	4 (3.5–4)	4 (4–5)	0.098
Did you have insomnia?	3 (1–3)	3 (3–4)	0.02	3 (2–4)	3 (2–4)	0.384
Do you experience extreme sadness sometimes?	2 (2–4)	3 (1–4)	0.417	2 (2–3)	2 (2–3)	0.384
Do you have financial concerns?	3 (3–4)	3 (3–5)	0.161	2 (2–3)	2 (1.5–3)	1
Do you tend to reduce the number of hospital visit?	3 (1–3)	3 (2–4)	0.094	3 (2–3.5)	3 (2–4)	0.271
Do you want to receive the cancer treatment at the hospital?	3 (1–3)	3 (1–3)	0.846	2 (1–3)	2 (1–3)	0.334
Do you not want to receive the cancer treatment?	2 (1–4)	3 (1–4)	0.527	3 (2–4)	3 (2.5–4.5)	0.582
Would you like to switch to a treatment that requires fewer visits?	3 (2.5–4)	3 (3–4)	0.574	3 (3–4.5)	3 (3–4)	0.582
Do you want to switch from intravenous regimens to oral regimens?	3 (2.5–3.5)	3 (2–3)	0.265	3 (2–3)	3 (2.5–3)	0.164
Do you want a longer drug holiday to reduce the number of hospital visits?	3 (2–3)	3 (2.25–3)	0.587	3 (2–4)	3 (2–3.5)	0.582
Do you want to receive chemotherapy that is effective but has stronger side effects?	3 (2–4)	4 (3–4)	0.294	4 (3–4)	4 (3–4)	0.855
Do you think it is acceptable to postpone a CT scan for 1–2 months?	3 (3–4)	3 (3–4)	1	3 (2–3)	3 (2–3)	0.334
Do you often look at our hospital’s website?	1 (1–2)	1 (1–2)	0.212	1 (1–2)	1 (1–2.5)	0.164
Do you feel that the hospital is taking sufficient measures against the spread of infections?	4 (4–5)	5 (4–5)	0.031	4 (3–4.5)	4 (3–4)	0.433

IQR: Interquartile Range.

## Data Availability

All databases are available from the corresponding author upon reasonable request.

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
