# Peer review of "Impact of the COVID-19 Pandemic on Patients with Gastrointestinal Cancer Undergoing Active Cancer Treatment in an Ambulatory Therapy Center: The Patients’ Perspective"

_healthcare, 2021, doi:10.3390/healthcare9121688_

Round 1

Reviewer 1 Report

The manuscript title is interesting however consider revising the manuscript to improve it. Consider using a checklist to guide the writing of this manuscript.

The introduction section needs major revision, it does not flow. For example, the discussion on cancer starts abruptly. 

The methods section is somewhat very limited in description which makes it difficult for another researcher to repeat the study. For example, no selection criteria indicated, how the questionnaire was developed, whether it was piloted before distribution, how were the questionnaires distributed? were they postal or self-administered questionnaires, face to face or via the telephone? the different variables were not described in this section nor was adequate description of the data analysis provided. no information on the sample size/power calculation has been provided. 

Results have been presented, however, the presentation can be improved. Since the sample size is small and without the power calculation it is not possible to make a proper conclusion. 

Referencing style is adequate.

Reviewer 2 Report

This is an overall hard-working research project, however, I didn't detect enough novelty, nor did I find it to be very interesting to the readers. I suggest the authors consider it in other lower-tier journals

Author Response

Thank you for your constructive comments.

We have reported that the COVID-19 pandemic has had a significant impact on the lifestyle and psychological wellbeing of patients with gastrointestinal cancer included in this study. However, the patients wished to continue their chemotherapy without any changes, according to the American Society of Clinical Oncology (ASCO) and the European Society of Medical Oncology (ESMO) guidelines. To the best of our knowledge, this is the first study to investigate the perspective of the patients on cancer treatment, specifically, of those with gastrointestinal cancer undergoing active chemotherapy before and during the COVID-19 pandemic. We believe that the findings of our study will be of interest to the readers in the healthcare field and that it fits the scope of the Healthcare in terms of health assessment and mental health. We have read the comments carefully and revised the manuscript. We hope that the revised manuscript will be approved for publication.

Reviewer 3 Report

The authors present the data from a patient survey to understand concerns of gastrointestinal cancer patients undergoing active treatment in an oncology centre in Japan. The topic is interesting and pertinent in the current pandemic, and I thank the authors for sharing their research

I would respectfully like to offer some comments/suggestions for the authors for consideration:

1. The abstract can be re-written to be more impactful and concise, for e.g. in under background in abstract "suit patients wishes".

2. In table 1, column titled N can be slightly misleading, and can be omitted or re-titled.

3. For the presentation of the survey data results, I would also encourage the authors to consider other methods of presentation of their data, aside from just median and IQR, especially so for questions on anxiety, insomnia and extreme sadness, an approach like a diverging stacked bar chart may be more visually convincing of the differences.

4. Methods also need to be further described to explain how these 18 questions were formulated or written and to explain if the pre-pandemic scores were truly obtained pre-pandemic or based on retrospective reporting (and if the latter then why a significant percentage of responses were not available).

5. Some English editing is also required (e.g. in abstract first sentence: "[severe] acute respiratory" severe being omitted; ASCO and ESMO abbreviations when written out in full should have the words appropriately capitalised; under methods "subgroup analyses by age and gender were performed" may be a more concise way of explaining the planned subgroup analysis, etc)

Thank you.

Round 2

Reviewer 1 Report

The manuscript is somewhat improved however there still grammatical errors-this requires further revisions in the English language. For example, consider revising the conclusion section just as the other sections. There is a contradiction within the paper on the sample size calculation. Please rephrase the statements on the power calculation in the section describing the limitations. Are you suggesting that the calculation was conducted retrospectively? Please clarify this in the manuscript.

Author Response

Thank you for your recognition of our work. 

We have proofread the manuscript again and uploaded the revised version. 

We have rewritten the sample size section and added a note that we did the sample size calculation retrospectively in limitation according to your suggestion.

Reviewer 2 Report

Although the authors spent time and worked hard on this study, I don't feel this one is interesting enough to the readers, and the novelty is not high. 

Author Response

Thank you for your constructive comments. 

We believe that the findings of our study will be of interest to the readers in the healthcare field and that it fits the scope of the Healthcare in terms of health assessment and mental health.